# CROSS ATTENTION TO EMBED ODDLY SHAPED DATA AND APPLICATIONS IN IONOSPHERIC MODELING

## ABSTRACT

It is desirable to have models of many physical phenomena, yet often data for these phenomena are oddly structured. These structures, such as ungridded and arbitrary length data prevents the use of many types of machine learning techniques, such as feed-forward neural networks. It is thus quite desirable to be able to move this data into a fixed size and shape for easier data ingest. We propose a method of using cross attention to do this. An example of oddly shaped data is Total Electron Content (TEC), or the vertical integral of electron density in the atmosphere. TEC data is calculated using both the position of a satellite and a position on the surface of Earth, giving a non-fixed location per sample. This leads to a splattering of points on the globe where measurements exist that change in shape and amount each time step. We apply our technique to TEC in an autoregressive approach. This allows us to both obtain an embedding describing the global TEC and create completed TEC maps, filling in where measurements are not taken. The global embedding can then be further used in other models.

## 1 INTRODUCTION

### 1.1 ODDLY SHAPED DATA

Oddly shaped data is data that does not present itself in a gridded or fixed-length form. Such data can vary in length from sample to sample and also can vary in shape (i.e., distribution) from sample to sample. An excellent example of this is Total Electron Content (TEC) data, which, because of the nature of computing it, does not have a set shape or structure (see Figure 1 for a visual on how TEC data changes shape at different timesteps). Additionally, during a single time step, TEC data is not ordered (i.e., to not introduce bias, the order data is processed should not affect the output).

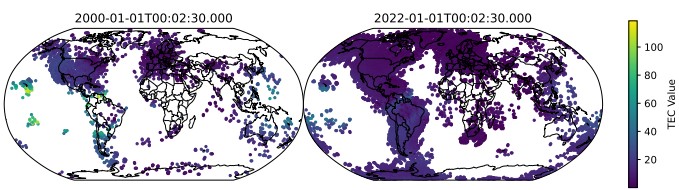

Figure 1: Example of TEC data showing variation in data shape between samples.

Because of the nature of oddly shaped data, it cannot be used in all applications. For example, such data cannot be ingested into feed-forward neural network models. This has led to the lack of using historical TEC data to drive electron density models, despite the advantages that would come with doing so (i.e., giving the model the integral of its output makes computing the output easier).

Despite the lack of using TEC data in electron density models, efforts have been made to "complete" TEC maps, or fill in the missing areas (Pan et al., 2021; Sun et al., 2023). Such a completed TEC map could, in theory, be used as an input to feed-forward neural network models. However, the act of completing data can introduce errors (Schafer & Graham, 2002), as the false data is biased by

the techniques used to create it. To avoid this, data can be used as-is, but often this requires some architecture or technique.

## 1.2 IONOSPHERE MODELING

The ionosphere impacts both ground and space assets and is strongly coupled with space weather. Of particular importance is the direct impact on wireless communications (Goodman & Aarons, 1990), especially satellite communications. This is because the ionosphere, which is the charged layer of the atmosphere, acts like a plasma and causes electromagnetic waves that pass through it to bend or even reflect. Such a strong impact on communications creates a desire to have an accurate model driven by space weather observations. Such a model would be important to GPS positioning (Dubey et al., 2006; Hoque & Jakowski, 2006), as well as other systems. Additionally, solar storms have can have impacts on power grids and electronics (Eastwood et al., 2017; Kappenman, 2010), as well as human health (Eastwood et al., 2017; Kiznys et al., 2020; Babayev & Allahverdiyeva, 2005). A representative measure of the impact of the ionosphere is the electron density, which directly impacts the aforementioned signal reflection. The electron density of the ionosphere is measurable, but such measurements are very localized, leaving us with a sparse measurement at any one point in time.

Electron density is measured in four main ways:

- GPS Radio Occultation (GPSRO), which analyzes signal paths that go through the ionosphere between GPSRO satellites and GPS satellites.
- Ionosondes, which send a variety of frequencies and measure return time.
- Incoherent Scattering Radar (ISR) (Gordon, 1958), which look at the intensity of the scattering of a return radar beam.
- In-situ measurements, such as Langmuir probes, which directly measure electron density in-place.

Electron density can also be indirectly measured by measuring the TEC, which is the integral of the electron density along the path between the satellite and receiver used. It is usually measured in the units of $TECU = 1 \times 10^{16} \frac{electrons}{m^2}$.

Additionally, there are many correlated or otherwise related measurements we can take of the Earth-Sun system. Many models use indices, which represent magnetic activity, and other common measurements.

### 1.2.1 EXISTING APPROACHES

Modeling the ionosphere is done in one of two main ways, either with closed-form physics-based modeling (Anderson et al., 1987; Huba & Krall, 2013; Fuller-Rowell et al., 1996; Scherliess et al., 2009) or empirical-based modeling (Bilitza, 2018; Themens et al., 2017), with both having advantages and drawbacks.

Physics-based models use known laws of physics, but need functions that are defined everywhere to predict specific outputs (i.e., they can only use directly modeled observations).

On the other hand, empirical models can be adjusted to incorporate additional factors, but are not necessarily restrained by physicals requirements.

Recently, Machine Learning (ML) approaches have become popular (Dutta & Cohen, 2022; Sai Gowtam & Tulasi Ram, 2017; Gowtam et al., 2019; Li et al., 2021; Chu et al., 2017; Habarulema et al., 2021). These models perform well compared to existing approaches, but they do not go much further than shallow feed-forward networks.

## 2 METHODOLOGY

### 2.1 TEC EMBEDDING

Our TEC data comes from the Madrigal database (W. Rideout & K. Cariglia).

Our data is split into training, validation, and test sets temporally. Training data is from 15-Oct-1998 to 31-Dec-2020, validation data is from 1-Jan-2021 to 31-Dec-2021, and testing data is from 1-Jan-2022 to 21-May-2023.

We aim to embed TEC data by using an autoencoder. This serves two purposes. First, it allows us to train a model to embed TEC data into a lower, fixed dimension without requiring additional data, and second, it has the side-effect of performing TEC completion, which, while not necessarily desirable for a model input, is useful as a science tool. We will later discuss the TEC completion, but it is not the primary goal of this TEC embedding.

We start our approach to embedding TEC data by following the structure for a transformer outlined by Vaswani et al. (2017). Our model is composed of two parts, the encoder and the decoder or reconstructor. For both we use the same architecture but change which inputs are the keys, values, and queries. For the encoder, we set the keys and values to be the input TEC observation sequences, which have been preprocessed to have their longitude represented by a sin/cos pair and latitude scaled to be between -1 and 1. We also normalize TEC. Thus, the observation sequences are scaled latitude, sine of scaled longitude, cosine of scaled longitude, and normalized TEC. This gives the inputs a shape of (batch_size, sequence_length, 4). Importantly, we omit the time (year, day of year, time of day) to force the model to learn the TEC values themselves instead of relying on using time. The query is a parameter of the encoder that acts like a learnable sequence. The multihead attention used is followed by a linear layer that changes from the internal embedding to a target size.

For the reconstructor, we treat the observation sequences as the queries and the embedding as the keys and values. There is a linear layer that goes from the embedded dimension to the attention embedding dimension. For both the embedder and reconstructor we use an internal attention embedding size that is the same.

We also look at chaining these embedding units together like transformer blocks, where the output sequence is used as the input query sequence instead at each step. We notice a lack of improvement by doing this, likely due to the experimental setup, so we do not present such models, but we plan to investigate the possibility of the multi-step embedding in future work.

A graphic of the architecture that we use may be see in Figure 2.

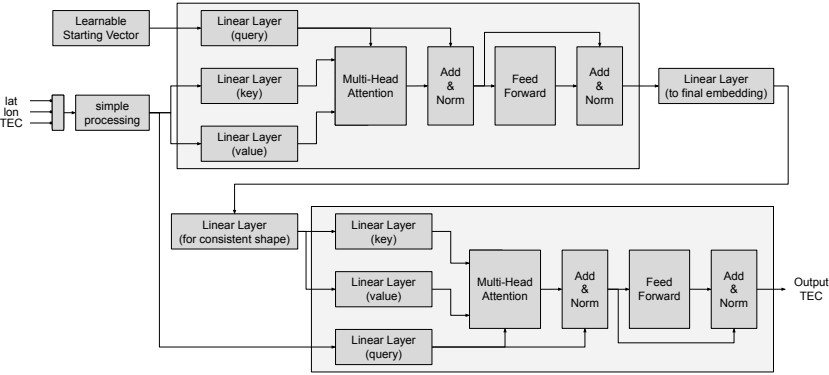

Figure 2: Architecture of autoencoder used to embed and reconstruct TEC data.

We train various versions of this autoencoder with different parameter sizes and compare their performance later in this paper. There we will note the size of various parameters. For all models, an internal attention size of 32 is used.

We also create two versions of a simple TEC predicting model. That is, we have a model that uses location, time, and driving indices (such as those used in electron density modeling that have been shown to be linked to changes in the electron density of the ionosphere) to predict TEC. One is a simple multi-layer feed-forward neural network and the other uses the same structure as the reconstructor used in our autoencoder. These models provide a comparison to the embedding model to demonstrate if more than just cyclical quiet-day information is being encoded. The latter performed much better and thus is the one used in comparisons.

## 2.2 IONOSPHERE MODELING

We have previously developed ML models for predicting electron density in the ionosphere. We use a simpler version of our model here to compare the benefits of using embedded TEC data to models without such information. These models aim to demonstrate that there is meaningful data captured in the TEC embeddings, and future work aims to improve upon them.

To use the embedded TEC data, we train the TEC model to embed a single timestep's set of TEC information into a vector of size 16 (we used the smaller sized embedding to save on memory and data storage). For each datapoint used in DINN, the most recent embedded TEC data is concatenated with the standard input vector. Any data where the most recent TEC information is greater than 5 minutes (TEC data is every 5 minutes) is rejected.

Our electron density models are trained on GPSRO data from the CHAMP (UCAR COSMIC Program, 2020a) and GRACE (UCAR COSMIC Program, 2020b) satellites and the COSMIC 1 (UCAR COSMIC Program, 2022) and COSMIC 2 (UCAR COSMIC Program, 2019) satellite constellations. Overall, this data spans approximately 100-600 km altitudes and 20+ years.

We pre-process to reject profiles with peak electron density outside of the 200-450 km altitude range with negative electron densities above 100 km (sub-100km negative densities are instead ignored), an approached introduced by Habarulema et al. (2021) to reject poor-quality samples.

Our data is split into training, validation, and test sets in a similar manner as the TEC data. Training data is from 11-Jun-2001 to 31-Dec-2020, validation data is from from 1-Jan-2021 to 31-Dec-2021, and testing data is 1-Jan-2022 to 5-Oct-2022.

For supplemental input data, we use Dst, Kp, ap, and f10.7 from NASA's OMNI hourly data (Papitashvili & King, 2020). Kp, ap, and Dst help describe the state of Earth's magnetic field, while f10.7 is the solar flux at a wavelength of 10.7cm.

For the ionosphere models, all models consist of input adjustments (e.g., sin and cos of $\frac{2\pi t}{24}$, where $t$ is the UT hour), similar to the autoencoder of the TEC model. These adjusted inputs are used as the base input tensor. This base input tensor is used in a simple feed-forward neural network we call Deeper Ionosphere Neural Network (DINN), as it has more layers (i.e., it is deeper) than previous feed-forward ionosphere neural networks. This architecture is visualized in Figure 3.

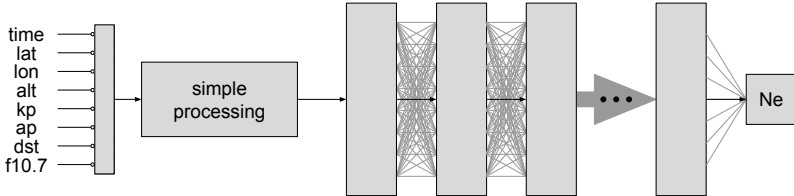

Figure 3: DINN architecture. This model uses feed-forward architecture with fully connected neuron layers, preceded by some simple input adjustment.

Standard model inputs are latitude, longitude, altitude, year, day of year, hour, Kp index, ap index, Dst index, and solar index f10.7. The inputs are adjusted before being used in the feed forward layers as follows: the latitude is scaled between -1 and 1, the longitude is turned into a sine/cosine pair of values. Altitude is scaled to be in megameters. The year is simply used to determine if there is a leap year so that the day of year can be turned into a sine/cosine pair. The hour of the day is also converted into a sine/cosine pair. The Kp index is divided by 10, the Dst and ap indices are divided by 20 and the f10.7 index is divided by 150.

The DINN architecture has an input layer size of 12 (28 when used with embedded TEC, as we have trained it with an embedding size of 16 for the TEC data), with hidden layers of sizes 64, 128, 128, 256, 128, 128, 64, and an output layer of size 1. The feedforward layers use Sigmoid activation functions. The model outputs the log of electron density.

For our models we use Mean Squared Error (MSE) on the log of electron density as our loss function. We train for a total of 100 epochs using a time-grouped batching approach and select the model where validation loss stops decreasing as the trained model. We use a time-grouped batching approach. We use a batch size of 16384 data points.

We also have more complex models, which we use for comparison. We have RDINN, which uses historical Kp, ap, Dst, and f10.7 values as inputs to RNNs whose subsequent inputs are concatenated like the TEC data is concatenated in DINN. We also have RDINN_XRS, which does the same but also with X-ray flux data. For the latter, GOES X-ray data is used as input data (NOAA), using all available satellites at the time of measurement and averaging satellites together when multiple were available. The historical index values are unscaled. We use hourly data of the past 3 days of observations for all four indices. We also include the past 81 days of daily-averaged f10.7 data. X-ray flux data is used for the past 90 minutes every minute, using both short and long wavelengths.

Each RNN section has a hidden size of 64 with 3 layers followed by a feedforward layer with an input size equal to 64 times the number of types of histories and output size of 16.

RDINN and RDINN_XRS are trained the same way as DINN, although because GOES XRS data is not always present, RDINN_XRS has seen fewer samples (1.28 billion vs 1.36 billion in each epoch).

## 3 RESULTS/DISCUSSION

### 3.1 TEC EMBEDDING

We traind a few versions of the autoencoder model. Final models were selected from lowest validation error after it stopped decreasing for more than approximately 5 steps. We have 3 versions of our autoencoder model, each with different embedding sizes. We outline the parameters for each of these in Table 1.

Table 1: Embedded TEC model parameters

| NAME | Embedding Sequence Length | Embedding Size |
|---|---|---|
| TEC_16 | 16 | 1 |
| TEC_64 | 16 | 4 |
| TEC_512 | 64 | 8 |

We also trained a decoder/reconstructor that has the same structure as the one used in TEC_512, except instead of encoding we simply use a linear layer to reshape the inputs of the model to what the reconstructor expects. The inputs to this are time, latitude, longitude, Kp, ap, Dst, and f10.7. Essentially, this model is a TEC predictor. We call this NoEmbedCompare, which we use as a comparison baseline to highlight if the model is capturing anything meaningful in the TEC data besides what can be reconstructed from a set of correlated driving parameters (these are the same as those used to predict electron density in our and other models).

Using these four models, we analyze the reconstruction error on our test data set. This may be seen in Figure 4. Here, we notice that as the size of the embedding increases so too does the accuracy of the reconstructed TEC. Notably, all embedding versions of the model perform noticably better than the simple prediction model, demonstrating that the embedding operation is capturing relevant information about the specific set of TEC values. From this, we can say that our technique is able to capture information from a sequence of arbitrary length and embed it to a fixed length.

Given that we are able to reconstruct from an embedding to an arbitrary sequence, it stands that we can also use this approach to complete TEC maps. While the accuracy of this completion is out of the scope of this paper (and we plan to do an analysis of this in the future), we present an example of TEC completion in Figure 5, where we can see a match with the expected shape of TEC, particularly the bifurcation along the magnetic field line. Such results are promising for future TEC completion work.

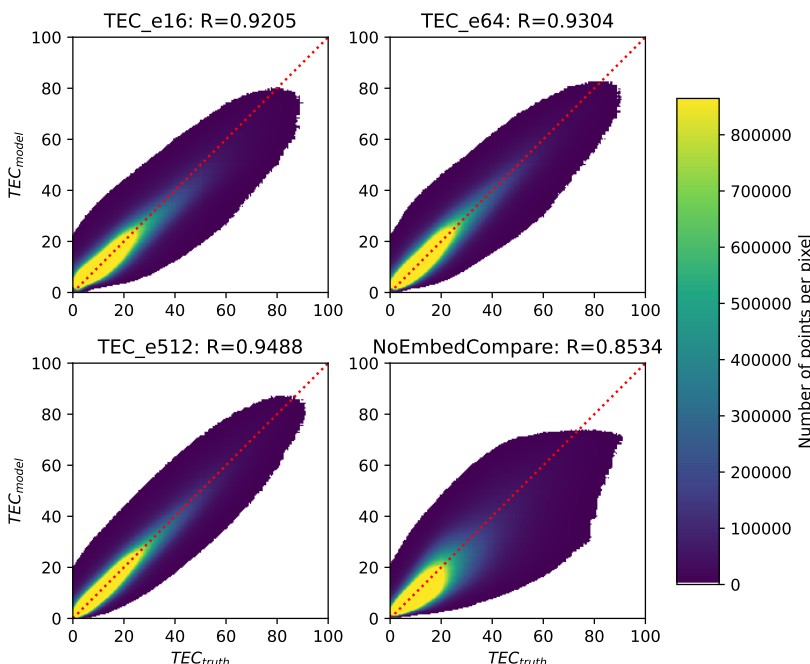

Figure 4: Correlation plots of reconstructed TEC vs testing data, with 2 billion test points. The ideal is represented as a straight line. Yellow data indicates concentrations above the colorbar limit. Included also is the Pearson correlation coefficient, with values closer to 1 representing a distribution closer to falling on the ideal line. Represented here are 3 versions of our autoencoder compared against a simple approach.

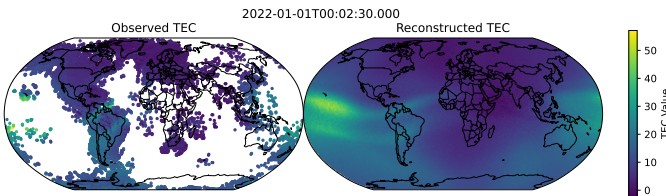

Figure 5: Example of TEC completion using our autoencoder approach. TEC_512 is used in this example.

From these tests, we demonstrate that our embedding technique is successful at embedding an arbitrary length sequence into a meaningful embedding, and that, as expected with larger embedding sizes, improvements can be seen when the embedding space is sufficiently large.

## 3.2 IONOSPHERE MODELING

We further use the embedded TEC to model the electron density of the ionosphere. In this case, we use TEC_16 for speed and ease of implementation, but acknowledge that model performance would likely increase with larger embeddings. We aim to do this with future work, but the goal of this paper is to present the embedding technique and demonstrate that it provides a functional way to use oddly shaped data, such as TEC. In this testing, we look at our electron density model, DINN, both with and without the size 16 embedding appended after the simple processing step. This gives us DINN_eTEC, which is DINN with the embedded TEC information.

We compare these two approaches using the GPSRO test data, as seen in Figure 6. It is clear that the DINN_TEC model is able to learn a better representation of GPSRO data and it has more accurate predictions.

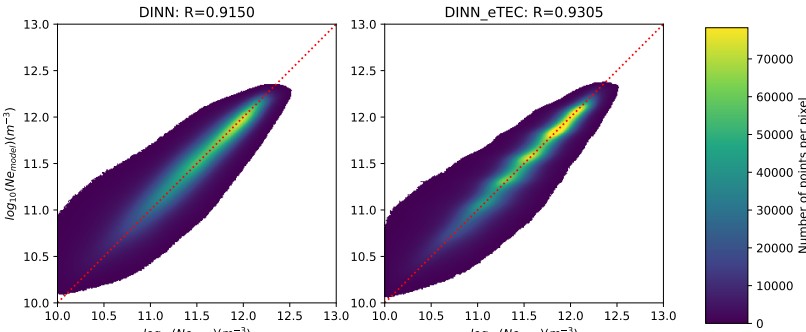

Figure 6: Correlation plots of electron density models vs GPSRO testing data, with 185 million test points. The ideal is represented as a straight line. Yellow data indicates concentrations above the colorbar limit. Included also is the Pearson correlation coefficient, with values closer to 1 representing a distribution closer to falling on the ideal line. Compared here is our DINN model with and without embedded TEC as an additional input.

We also consider ionosonde data, particularly hmf2 and nmf2, or the height and electron density at the peak of the f2 layer of the ionosphere, respectively. The ionosonde data comes from the Global Ionospheric Radio Observatory (GIRO) (Reinisch & Galkin, 2011). We consider high Kp, or storm-time data, with $Kp * 10 \geq 50$. The Pearson correlation coefficent as well as the mean absolute error for this is reported in Figures 7 and 8.

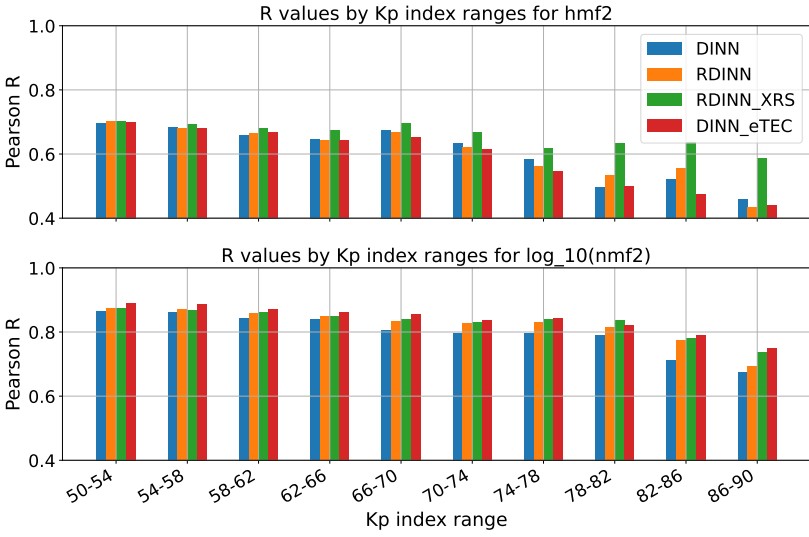

Figure 7: Correlation of the predictions against ionosonde measurements along Kp ranges. Higher is better.

This test demonstrates that the model is able to predict well the peak electron density but not the altitude of the peak electron density. Specifically, we see a decrease at all Kp levels for mean absolute error and an increase in Pearson correlation coefficient when considering nmf2 and comparing DINN

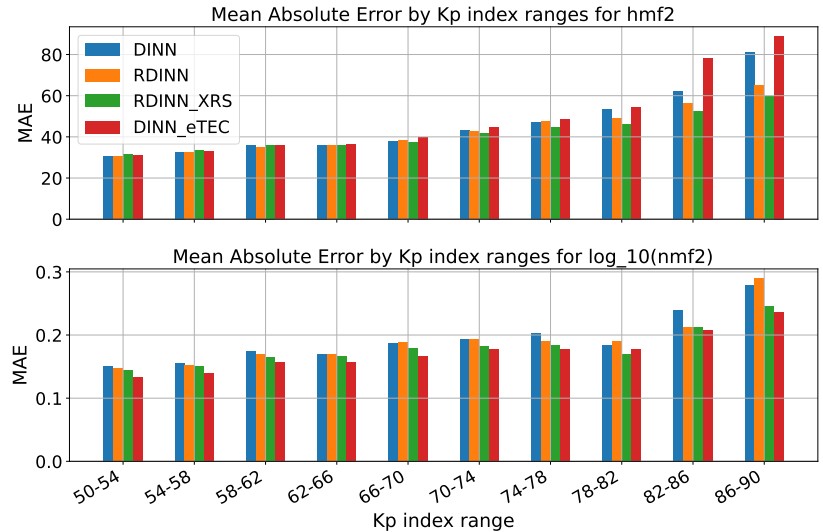

Figure 8: Mean absolute error of the predictions against ionosonde measurements along Kp ranges. Lower is better.

vs DINN_eTEC. We do not see such a trend for hmf2, where often DINN_eTEC performs worse (lower correlation and higher error). We believe that this is because the TEC data has enabled the model to overfit to GPSRO data (we see an odd pattern emerging in Figure 6), as GPSRO data is a product of a slanted TEC measurement and the electron density derived from it is biased by the underlying model used. Additionally, TEC is not as directly correlated with the shape of the 3D electron density profile, but it is strongly correlated with the value of the peak electron density value. Thus, it is reasonable that such a model would perform well in terms of nmf2 predictions but the addition of TEC would not necessarily improve hmf2 predictions.

We include here also results from other versions of our models. We include these because we want to highlight two points. First, that we have lower nmf2 error (and higher correlation) for predictions using the TEC embedding version, even when compared to the more complex models, and second, that hmf2 error can be reduced by including more information about various driving factors of ionization, such as X-ray flux, whose information is not directly captured in historical TEC information. Thus, we have a situation where we demonstrate the limitations of using just embedded TEC information to drive storm predictions while also highlighting its impact in predicting the peaks of electron density in storm predictions. Further work aims to use embedded TEC information alongside X-ray flux to provide even better ionosphere modeling.

## 4 CONCLUSIONS AND FUTURE WORK

In this paper we proposed a technique to take arbitrarily shaped data and embed it to fixed-length vectors. In particular, we used TEC data, although there are other types of appropriate data that it could be used on, such as weather data. We demonstrated that our technique is able to encode essential information about the sequences, as reconstruction is improved against the lack of an embedding step. Additionally, using larger embeddings increases performance.

Our embedding technique also has the extra benefit of being able to provide completed TEC maps. We aim to further pursue this in future work, but highlight within this paper that the ability to produce reasonable looking TEC maps hints towards the technique's ability at encoding information.

After demonstrating that information can be successfully encoded using our transformer-like technique, we then demonstrated an example use of embedded arbitrary sequences. By including TEC information in electron density prediction models we see an increase in model performance, particularly in regards to the electron density itself, while we note drawbacks in increasing electron density

profile accuracy. We assume that this is due to the loss of vertical information when integrating (i.e., when measuring TEC the shape of the ionosphere is not captured). We aim to couple the TEC version of our model with other inputs to make up for the lack vertical information provided by the TEC in the future.

Overall, our paper aims to demonstrate the feasibility of a technique to encode oddly shaped data and present an example of how such an embedding may be used.

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
