# OpenReview forum: "Cross Attention for Oddly Shaped Data and Applications in Ionospheric Modeling"
_ICLR.cc/2025/Conference — ICLR 2025 Conference Withdrawn Submission_

### Official Review · Reviewer_Js79 · 2024-10-20

**Soundness:** 1
**Presentation:** 1
**Contribution:** 1
**Rating:** 1
**Confidence:** 5

**Summary:**

The paper describes an autoencoder with attention layers used to describe ionospheric data. The preprocessing of the data is described, e.g. scaling latitude from -1 to 1, and longitude as sine cosine pair, and feature like the Kp index are used. I did not find a description of the Kp index anywhere even though it appears to be a significant value whose predictions are given in a plot.  Can you please provide a description of the Kp-index, ap-index and the rest of the technical features?
The were companions of different models, changing only with respect to their inputs, and a paragraph describing using the outputs as inputs suggesting that some form of recurrent processing was also attempted but did not yield results. I did not see a description of a second machine learning model nor a comparison with another model performing this task.

It might be that there is a community mismatch with this paper because the paper only describes the use of a particular autoencoder on the data and it is not compared with an alternative approach for the same models. Comparisons with “different models” are there only in terms of different inputs and not different ML algorithms. A substantial problem for the publication of this work in a Machine Learning conference is also that the type of input is not clearly introduced, terms such as the Kp index, ap index, and solar index are not clearly defined. I guess the authors think that a reference is enough for the reader to understand the data, but since the paper focuses on using different combinations of the data to demonstrate performance discrepancies then the data should be explained in more detail. It is uncommon for a method to be described without a comparison to a different method in a machine learning conference. Such a comparison would be useful here. Perhaps trying to embed your data to images and using a CNN or ViT-based model would give you another approach to compare against.
Also, submissions to ICLR are expected to be reproducible, however, since the authors do not provide their code, all implementation details should be described in the paper, including size of different layers and learning rates.

The paragraph in line 131 implies that the authors tried to create some form of recurrent algorithm by taking the output of the autoencoder and using it as input to the same autoencoder. This is not clearly described but it appears like a technically weak attempt at creating a recurrent algorithm for the data. I would suggest a deeper exploration of the literature on that.


As a general comment, I would suggest that the authors present a deeper analysis of their model with more thorough comparisons to alternative approaches.

**Strengths:**

The paper describes a novel application of an autoencoder. The problem is uncommon and results of some resasonable quality were achieved.

**Weaknesses:**

The method is not clearly described, it is unlikely that a reader can reproduce that work from the available material. There is no comparison with another work. Since the application is novel (at least I have not seen an ionospheric model with autoencoders before) it would be appropriate to describe the problem better. Variable names are introduced but they are not explained in the text, e.g. Kp index, ap index etc.

**Questions:**

The title of the paper mention oddly shaped data, can you provide a defintion of the term oddly shaped data?
Can you please give some more examples of oddly shaped data?
Are there any other methods for describing oddly shaped data that you could compare with your approach?
What is Kp index, ap index, Dst, etc. ?
Which optimisation algorithm, learning rate, and other hyperparameters did you use for your experiments?
Are there other types of oddly shaped data one could apply your method to?

---

### Official Review · Reviewer_Vnbn · 2024-10-30

**Soundness:** 3
**Presentation:** 3
**Contribution:** 2
**Rating:** 3
**Confidence:** 3

**Summary:**

The paper proposes a method using cross-attention within an autoencoder to embed and reconstruct oddly shaped data, specifically Total Electron Content (TEC) data for ionospheric modeling.

**Strengths:**

- Important problem setting, well motivated
- Proper language and structure of the paper
- Good and thorough experimentation  section

**Weaknesses:**

- No methodological innovation. Applying an existing approach to a new dataset unfortunately  does not pass the bar for methodological contributions
- There are hints of controlling for confounding factors that would bias the representation and the downstream prediction but it is not thoroughly explored. I recommend that this is further investigated
- Insufficient literature review with many ML <> Ionospheric physics papers missing

**Questions:**

- What are the main confounding factors besides the time and place of the measurements that would affect the performance of the model ?


Overall this is a solid paper for an ionospheric physics journal or conference , not an ML one where methodological innovation and contributions are key

---

### Official Review · Reviewer_3bFB · 2024-11-03

**Soundness:** 3
**Presentation:** 3
**Contribution:** 2
**Rating:** 3
**Confidence:** 4

**Summary:**

The paper proposes the notion of “oddly shaped data” with the example of Total Electron Content (TEC) data, and propose and embedding to describe the global TEC and create completed TEC maps.

**Strengths:**

The notion of TEC and ionospheric modeling is interesting.

**Weaknesses:**

The motivation is unclear, particularly the notion of “oddly shaped data” and the contribution is unclear. The authors mention “unguided and arbitrary length”, “a splattering of points on the globe where measurements exist that change in shape and amount each time step”, but it seems data is being gridded across the surface of the earth, and the main problem appears to be sparse sampling.

There is no clear mathematical description of the data or method used, other than a block diagram in Figure 2 which appears to be mostly common components.

**Questions:**

How is the idea here of "oddly shaped data" here different from "missing data", which is solved with many different approaches starting with expectation maximization?

---

### Official Review · Reviewer_4vsu · 2024-11-04

**Soundness:** 1
**Presentation:** 1
**Contribution:** 1
**Rating:** 1
**Confidence:** 4

**Summary:**

The authors propose an autoencoder architecture for ionospheric total electron count data based on transformers. They utilize the obtained lower-resolution representation to construct a regression model to predict the electron density in the atmosphere. This data is temporal in nature, and geographically sparse with varying available locations per time slice.

**Strengths:**

The paper considers an interesting and relevant application domain, i.e. the domain of predicting electron density in the ionosphere.

**Weaknesses:**

In my opinion, there are several problems with this submission in the areas of presentation, methodology, and comparison to competing methods.

First, the claim of introducing a new class of ‘oddly-shaped data' seems fuzzy and unwarranted to me. The TEC data presented appears to be simply a sequence of images with missing values, akin to some inpainting tasks or inputs for masked autoencoders, where the goal is to fill in / infer the missing values in said images (reference examples provided below).

Second, I don't believe the decision to turn this data into sequences and apply models designed for natural language processing is valid here. There is no true order among the set of pixels for which data is available in any given time slice, because the data is indexed by 2 dimensions. Therefore, turning it into a 1D sequence must be inevitably completely arbitrary. If the positions of elements have no meaning and are random, the transformers and their attention heads cannot learn anything meaningful about co-occurrences of positions. I think this data would be better served by image-based approaches, i.e. same-shape (World-coverage) images with missing/masked values, which can then be input to e.g. a masked autoencoder. See reference examples below.

Some work on masked autoencoders and inpainting worth looking at include:
1. Deepak Pathak, Philipp Krahenbuhl, Jeff Donahue, Trevor Darrell, and Alexei A Efros. Context encoders: Feature learning by inpainting. In Proceedings of the IEEE conference on computer vision and pattern recognition, pp. 2536–2544, 2016. URL https://arxiv.org/pdf/ 1604.07379
2. Kaiming He, Xinlei Chen, Saining Xie, Yanghao Li, Piotr Dollár, and Ross Girshick. Masked autoencoders are scalable vision learners, 2021. URL http://arxiv.org/abs/2111.06377
3. Zhaowen Li, Zhiyang Chen, Fan Yang, Wei Li, Yousong Zhu, Chaoyang Zhao, Rui Deng, Liwei Wu, Rui Zhao, Ming Tang, and Jinqiao Wang. MST: Masked self-supervised transformer for visual representation, 2021. URL http://arxiv.org/abs/2106.05656
 4. Ioannis Kakogeorgiou, Spyros Gidaris, Bill Psomas, Yannis Avrithis, Andrei Bursuc, Konstantinos Karantzalos, and Nikos Komodakis. What to hide from your students: Attention-guided masked image modeling. In Shai Avidan, Gabriel Brostow, Moustapha Cissé, Giovanni Maria Farinella, and Tal Hassner (eds.), Computer Vision – ECCV 2022, pp. 300–318. Springer Nature Switzer-land, 2022. ISBN 978-3-031-20056-4. doi: 10.1007/978-3-031-20056-4 18. URL https://link.springer.com/chapter/10.1007/978-3-031-20056-4_18

Third, I think the presentation of prior/related work on the topic is done is a somewhat misleading manner. Specifically, in lines 99-10 we read:

"Recently, Machine Learning (ML) approaches have become popular (Dutta & Cohen, 2022; Sai Gowtam & Tulasi Ram, 2017; Gowtam et al., 2019; Li et al., 2021; Chu et al., 2017; Habarulema et al., 2021). These models perform well compared to existing approaches, but they do not go much further than shallow feed-forward networks."

However, the authors dismiss the most relevant image-based approaches such as:
1. "TEC Map Completion Through a Deep Learning Model: SNP-GAN" (Pan et al, 2021)
2. "Complete Global Total Electron Content Map Dataset based on a Video Imputation Algorithm VISTA" (Sun et al., 2022)

by claiming that "the act of completing data can introduce errors (Schafer & Graham, 2002), as the false data is biased by the techniques used to create it." (line 53).

I don't think it's fair to claim that modern GAN and other image ML-based techniques bias the result more than what the authors propose, especially given the arbitrary transition to sequence data from pixels. Moreover, most GANs also work in a low dimensional latent space. That means that a trained GAN's decoder model can be inverted or even trained to include an inverse mapping from image space back to the latent space. This would allow to obtain latent codes also from GANs and directly compare their usefulness for the task at hand (i.e. regression of electron density). This was not done and the authors instead only compared their embedding-enabled regression model to very simple baselines. Please consider conducting additional experiments to compare GAN latent codes to the proposed method's embeddings.

Fourth, it is unclear to me if the presented results indicate substantial improvement. Figure 6 shows an increase of the $R^2$ measure from $0.915$ to $0.93$. Figure 7 also does not show significant correlation improvements for the authors' "DINN_eTEC" method. Figure 8. Top does not show significant MAE reduction for "hmf2". What magnitude of changes are considered significant in the context of ionospheric modeling?

**Questions:**

I think some important details are missing from the autoencoder architecture. For example:

- What is the "learnable starting vector" shown in Fig. 2? I could not find any reference to it in the text
- How is the latent space constructed? Is there some kind of random generator for the latent embedding values? Or maybe a deterministic scheme?
- How exactly is the autoencoder trained? What is the loss function? Does it predict all channels of the sequence? (i.e. is it also measured on how well it reconstructs e.g. the lon/lat values)

Regarding the authors' claim: "Given that we are able to reconstruct from an embedding to an arbitrary sequence, it stands that we can also use this approach to complete TEC maps. "
I'm wondering how this can be done exactly, because the decoder ("reconstructor") must also consume a latent embedding value alongside the lon/lat coordinates. So how would the authors derive these latent codes for the pixels with missing values? Is it somehow by random sampling? If so, how can the authors ensure that the values "make sense" from a physical perspective?

Did the authors consider using an image-based approach with missing/masked values? (sample references provided above)

Did the authors consider utilizing an existing GAN based method, e.g. cited in line 52, "TEC Map Completion Through a Deep Learning Model: SNP-GAN" and doing GAN inversion to obtain latent codes for ‘true' pixels (with available values) and comparing the latent codes from the GAN with latent codes from their transformer based autoencoder on the downstream task of regression?

---

### Note · Authors · 2025-01-17

I have read and agree with the venue's withdrawal policy on behalf of myself and my co-authors.